# Long-Term Effects of Hospitalization for COVID-19 on Frailty and Quality of Life in Older Adults ≥80 Years

**DOI:** 10.3390/jcm11195787

**Published:** 2022-09-29

**Authors:** Marcello Covino, Andrea Russo, Sara Salini, Giuseppe De Matteis, Benedetta Simeoni, Flavia Pirone, Claudia Massaro, Carla Recupero, Francesco Landi, Antonio Gasbarrini, Francesco Franceschi

**Affiliations:** 1Emergency Department, Fondazione Policlinico Universitario A. Gemelli, Istituti di Ricovero e Cura a Carattere Scientifico (IRCCS), 00168 Rome, Italy; 2Faculty of Medicine, Università Cattolica del Sacro Cuore, 20123 Milano, Italy; 3Geriatrics Department, Fondazione Policlinico Universitario A. Gemelli, Istituti di Ricovero e Cura a Carattere Scientifico (IRCCS), 00168 Rome, Italy; 4Department of Internal Medicine, Fondazione Policlinico Universitario A. Gemelli, Istituti di Ricovero e Cura a Carattere Scientifico (IRCCS), 00168 Rome, Italy; 5Department of Internal Medicine and Gastroenterology, Fondazione Policlinico Universitario A. Gemelli, Istituti di Ricovero e Cura a Carattere Scientifico (IRCCS), 00168 Rome, Italy

**Keywords:** COVID-19, clinical frailty scale, 5L-EQ-5D, older adults, age ≥80 years

## Abstract

**Objectives:** This study aimed to assess the effects of frailty and the perceived quality of life (QOL) on the long-term survival (at least 1 year) of patients ≥ 80 years hospitalized for COVID-19 and the predictors of frailty and QOL deterioration in survivors. **Design:** This is a single-center, prospective observational cohort study. **Setting and Participants:** The study was conducted in a teaching hospital and enrolled all COVID-19 patients ≥80 years old consecutively hospitalized between April 2020 and March 2021. **Methods:** Clinical variables assessed in the Emergency Department (ED), and during hospitalization, were evaluated for association with all-cause death at a follow-up. Frailty was assessed by the clinical frailty scale (CFS), and the QOL was assessed by the five-level EuroQol EQ-5d tool. Multivariate Cox regression analyses and logistic regression analyses were used to identify independent factors for poor outcomes. **Results:** A total of 368 patients aged ≥80 years survived the index hospitalization (age 85 years [interquartile range 82–89]; males 163 (44.3%)). Compared to non-frail patients (CFS 1–3), patients with CFS 4–6 and patients with CFS 7–9 had an increased risk of death (hazard ratio 6.75 [1.51, 30.2] and HR 3.55 [2.20, 5.78], respectively). In patients alive at the 1-year follow-up, the baseline QOL was an independent predictor of an increase in frailty (OR 1.12 [1.01, 1.24]). Male sex was associated with lower odds of QOL worsening (OR 0.61 [0.35, 1.07]). **Conclusions and Implications:** In older adults ≥80 years hospitalized for COVID-19, the frailty assessment by the CFS could effectively stratify the risk of long-term death after discharge. In survivors, the hospitalization could produce a long-term worsening in frailty, particularly in patients with a pre-existing reduced baseline QOL. A long-term reduction in the perceived QOL is frequent in ≥80 survivors, and the effect appears more pronounced in female patients.

## 1. Impact Statement

We certify that this work is novel clinical research, prospectively evaluating, for the first time, the long-term effects of COVID-19 hospitalization on adults ≥80 years. The research explores both the factors associated with the survival of discharged patients and the long-term effect on frailty and quality of life in the survivors.

### 1.1. Key Points

In patients ≥80 years the stratification of frailty by the Clinical Frailty Scale (CFS) could predict the long-term survival after hospitalization for COVID-19.

Hospitalization could produce a long-term worsening in frailty itself by triggering a self-feeding mechanism between the increased frailty and the increase in the mortality risk in case of a new infection.

A long-term reduction in the perceived QOL could be expected in the majority of older COVID-19 survivors.

### 1.2. Why This Matters

The risk stratification by frailty could suggest strict surveillance of the frailest survivors. Female patients with persisting symptoms could be the object of specific follow-up strategies and geriatric interventions to limit the long-term deterioration of their quality of life.

## 2. Introduction

Since December 2019, the world has been plagued by COVID-19 [1,2]. Although vaccination campaigns have started in most countries, the number of affected patients and the death toll is still increasing [3].

Italy faced one of the worst clusters of COVID-19, and mortality was particularly high [4]. The high median age of the Italian population was one of the main causes of this, with patients ≥80 years old being the most at risk of death caused by COVID-19 [5,6,7,8,9,10,11]. Most of the current research focuses on the presence of multiple comorbidities in older adults to explain the disproportionate death rate of these patients [1,2,5,6,7,8,9,10,11,12]. However, it was evidenced that comorbidity alone cannot comprehensively predict the extremely poor outcomes observed in older COVID-19 patients [13].

Older adults have highly heterogeneous baseline clinical conditions. Chronological age and comorbidities alone do not always reflect the overall health status of older patients. The concept of frailty was introduced to include several dimensions of physical fitness and autonomy, and it describes the progressively declined physiologic function and diminished strength leading to vulnerability and reduced resilience to stressors [14]. Frailty is demonstrated to be an independent predictor of poor outcomes in hospitalized patients with several clinical conditions as well as COVID-19 [13,14,15,16,17,18,19].

Given the immense burden that hospitalization for COVID-19 places on older individuals, it is obvious to expect long-term effects on the health status, frailty, and overall quality of life (QOL) of older COVID-19 survivors. The dramatic impact of severe COVID-19 on QOL and overall frailty has already been shown [20,21]. However, no data are available for the long-term follow-up of patients ≥80 years, who are indeed expected to have worse outcomes.

This study assesses a long-term (1 year) prospective follow-up of patients ≥80 years to determine the factors affecting overall mortality and the factors associated with an increase in frailty, as well as the overall reduction in the QOL of survivors.

## 3. Methods

### 3.1. Study Design

This is a single-center, prospective observational cohort study, conducted in an urban teaching hospital, which is a referral center for COVID-19 in central Italy. The ED has an annual attendance of about 75,000 patients and serves as a tertiary referral center for an area of 1.8 million inhabitants. According to the Italian registry data, about 7.2% of the residents in the area are ≥80 years old, and they represent about 21% of those accessing the ED in our institution.

The study enrolled all the patients ≥80 years old who were consecutively admitted to our ED from April 2020 to March 2021 and subsequently hospitalized. The diagnosis of COVID-19 was confirmed based on the WHO interim guidance and a positive result on a real-time reverse transcriptase–polymerase chain reaction assay of nasal and pharyngeal swab specimens [22].

Patients that did not receive complete frailty and QOL assessments in the ED, and patients who refused to participate in the study, were excluded from the analysis. The follow-up was assessed from the time of index ED admission for COVID-19.

### 3.2. Study Variables

All patients were assessed in the ED to retrieve the following clinical and demographic data:Age and gender.Overall frailty as assessed by the Clinical Frailty Scale (CSF) [23]. According to the CSF scale, patients were further categorized as fit for CSF 1–3 (corresponding to fit and mild vulnerability), vulnerable for CSF 4–6 (corresponding to vulnerable or mild frail), and frail for CSF 7–9 (corresponding to moderate to severe frailty).Quality of life, assessed based on the five-level EUROQOL questionnaire (EQ-5D-5L) [24]. The EQ-5D-5L is a standardized measure of health status validated to provide a simple and reproducible generic measure of QOL. We considered, in our analysis, the crude sum of all the points (best (1) to worst (5)) assigned to the five domains ascertained: mobility, self-care, usual activities, pain/discomfort, and anxiety/depression.Dependency on activities of daily life (ADL) based on the clinical status before the SARS-CoV-2 infection.Delirium occurrence, assessed based on the Richmond Agitation–Sedation Scale [25] during the first 24 h of ED admission.Physiological parameters, including body temperature, heart rate, respiratory rate, blood pressure, Glasgow Coma Scale, and peripheral oxygen saturation. Based on these measures, the NEWS score was calculated for each patient [26].The need for mechanical ventilation (MV), defined as the need for MV including non-invasive techniques and high-flow oxygen therapy for more than 24 h.Clinical history and comorbidities, including cognitive impairment, assessed based on the Charlson Comorbidity Index (CCI) for each patient [27], calculated at the time of index ED access.A laboratory evaluation and a blood gas determination in the ED of all patients. The values considered in the study were the first values obtained at ED admission.The length of hospital stay of the index admission, calculated from ED access to death or hospital discharge. The overall follow-up was calculated from ED access to the last follow-up assessment or death.The number of persisting post-COVID symptoms, ascertained by a standard questionnaire, including fatigue, dyspnea, joint pain/myalgia, chest pain, cough/sputum, anosmia/dysgeusia, sore throat, and diarrhea. The symptoms were included in the count if present for at least 1 month and not present before hospitalization.

### 3.3. Study Endpoints

The primary study endpoint was the all-cause death at the follow-up. As secondary endpoints, we evaluated the worsening in frailty defined as an increase of at least 1 point in the clinical frailty scale, and the worsening in QOL, defined by an increase in at least 1 point in the total value of the EQ-5D-5L. Both the changes in frailty and QOL were assessed at least 1 year from the index admission of the surviving patients.

### 3.4. Statistical Analysis

Continuous variables were reported as median [interquartile range] values and were compared using univariate analysis by the Mann–Whitney U test or the Kruskal–Wallis test in case of three or more groups. The categorical variables were reported as absolute numbers (percentage) and were compared by the chi-square test (with Fisher’s test if appropriate).

The follow-up and length of hospital stay were calculated from the time of ED admission to the last medical assessment or death. Survival curves were estimated by the Kaplan–Meier method. The study variables were assessed for association with all-cause death by a univariate Cox regression analysis. The significant variables in the univariate analysis were entered into a multivariate Cox regression model to identify independent risk factors for survival. As the analysis focused on patients ≥80 years, the Cox regression model was adjusted for the expected survival at one year for each 5-year step of age. The data on expected mortality were based on the Italian national registry of population, as assessed in 2019, to avoid the effect of the COVID-19 pandemic on expected survival [28]. To avoid model redundancy or overfitting, single items composing derived variables (CCI, NEWS, and dependency on ADL, which was included in the EQ-5D-5L) were excluded from the multivariate analyses. Some of the continuous variables (NEWS and CCI) were categorized into dichotomous parameters to consent to the direct evaluation of the outcomes compared with the original dataset^16^. The multivariate association of factors with the risk of all-cause death was expressed by a hazard ratio (HR) [95% confidence interval].

Factors significantly associated in the univariate analysis with the increase in the CFS and EQ-5D-5L at one year were evaluated in a multivariate logistic regression model to identify the independent predictors of each defined outcome. The association with these two endpoints was expressed by an odds ratio [95% confidence interval. A two-sided *p* ≤ 0.05 was set for significance in all analyses. Data were analyzed by SPSS v25^®^ (IBM, Armonk, NY, USA).

### 3.5. Statement of Ethics

This study was conducted following the Declaration of Helsinki and its later amendments, and it was approved by the local Institutional Review Board (IRB #001705520). The included patients gave informed consent to be included in the analysis.

## 4. Results

### 4.1. Study Cohort and Baseline Characteristics

In the period of 1 April 2020 to 31 March 2021, 843 patients ≥80 years positive for COVID-19 were evaluated in our ED. Among them, 729 were admitted to the hospital, and they constituted the original cohort included in the index COVID-19 hospitalization. The enrolled patients had a median age of 85 years [82, 89] and there were 346 males (47.3%). Overall, 441 (60.5%) patients survived the index hospitalization. These patients had a median age of 85 years [82, 88], and 193 (43.8%) were males. Overall, 73 (16.5%) patients were lost at the follow-up, leaving 368 patients in the study cohort. The median follow-up of these patients was 15 months [6, 18].

### 4.2. Factors Associated with Long-Term All-Cause Death after Hospital Discharge

Among the 368 patients who had full follow-up data, we recorded 132 (35.8%) deaths. Deceased patients were significantly older, and there were more females in this group (Table 1). Most of the deceased patients deceased a few months after the index hospitalization, and the median follow-up of the deceased was 4 months [2, 6]. Interestingly, all the deaths recorded in this group were within one year of the index hospitalization.

Not unexpectedly, the surviving patients were less frail according to the CFS, and the crude overall mortality was 3.8% in CFS group 1–3, 30.4% in CFS group 4–6, and 73.4% in the frailest group (CFS 7–9). Interestingly, the mortality for each CFS group was significantly different even, after adjustments for age, expected survival, sex, comorbidities, and relevant clinical factors during the hospitalization (Table 1, Figure 1).

Some of the clinical characteristics of the COVID-19 disease at the index hospitalization were associated with poor outcomes in the univariate analysis: particularly the severity of the disease at admission (as assessed by a NEWS > 5), the occurrence of delirium, and the overall length of hospital stay. However, when these factors were adjusted for frailty, age, and expected mortality, they did not result in an effect on the long-term survival of the patients (Table 1). 

### 4.3. Factors Associated with a Worsening in Frailty Status at One Year

Overall, 236 patients survived and could be evaluated for frailty changes at the follow-up. About two in five of the survivors (87/226, 38.5%) experienced a worsening in frailty, as estimated by the CFS (Table 2, Figure 2).

Interestingly, most of the considered factors did not show a significant association with an increase in frailty. This includes factors associated with the characteristics of the index hospitalization, with the possible exception of the length of the hospitalization. This, however, did not reach statistical significance.

Female sex and the QOL before COVID-19 emerged as possible predictors for an increase in frailty; however, after the multivariate adjustment, only the QOL before COVID-19 (as expressed by the EQ-5D-5L value) was independently associated with the risk of an increase in frailty (HR 1.12 [1.01, 1.24], *p* = 0.027) (Table 2).

Patients with increased frailty had more persistent covid symptoms compared to the controls, and a concurrent reduction in overall QOL (Table 1).

### 4.4. Factors Associated with a Worsening in the QOL Status at One Year

The overall QOL of these survivors ≥80 years was dramatically affected by the hospitalization for COVID-19, as more than half of the patients experienced a decrease in overall EQ-5D-5L value (136/226, 60.2%) (Table 3, Figure 2).

Factors most influencing the decrease in QOL were found to be the female sex, frailty status before COVID-19, age group, and overall pre-existing EQ-5D-5L value.

After adjusting for significant covariates, only the female sex emerged as a relevant risk factor for a decrease in QOL, with the odds for the male sex being about 30% lower compared to females (Table 3). Interestingly, persistent COVID-19 symptoms, a lower CFS, and a higher number of hospital re-admission were observed in the worsened QOL group, compared to the controls (Table 3).

## 5. Discussion

The main finding of the present study is that, in patients ≥80 years surviving hospitalization for COVID-19, the frailty assessment from the index admission could accurately recognize patients at an increased risk of long-term all-cause death. The frailty evaluation could recognize the patients at most risk independently from other relevant clinical factors, such as the severity of the disease, the need for MV, the length of the hospital stay, and comorbidities. Other relevant findings include the dramatic effect on QOL and the increase in frailty following COVID-19 hospitalization, which is independent of the age group and appears to affect the female sex more severely.

The effect of COVID-19 hospitalization on the functional decline and quality of life of surviving patients was already evidenced by some researchers [29,30,31,32,33]. However, all these studies were based on a shorter follow-up or younger cohorts, limiting the comparability of the obtained data on patients ≥80 years.

On one hand, younger patients have a baseline higher chance of survival from the hospitalization for COVID-19, thus leading to the possibility of a higher number of patients surviving with a persistent physical and psychological disability after hospital discharge [33]. On the other hand, younger patients could be expected to have a better baseline physical condition and a prompt recovery after the disease, as shown by the CFS, ranging between 1 and 2 in the 47 patient cohort reported by Carenzo et al. [34]. Moreover, as shown by Huang et al. [30] and by Vlake et al. [33], most of the patients experienced a progressive increase in QOL from the first months after discharge to the later follow-up, limiting the effective utility of a short-term assessment of QOL on the evaluation of persistent COVID-19 sequelae. Most of all, because patients ≥80 years had the worst survivability to COVID-19 hospitalization in most of the study cohorts [6,7,8,9,10,11,12,34], we can expect a “harvest” effect in selecting the most healthy individuals, regardless of the clinical and demographic factors considered. As a result of the above considerations, the actual long-term effect on patients ≥80 years could hardly be extrapolated from studies not specifically conducted on this population. As a final clue to this point, it must also be acknowledged that in many countries’ reports and in most research papers, older adults are often considered in a single category ≥ 65 years, further limiting the possibility of evaluating the effects of the pandemic on older groups of patients in their 8th and 9th decade. 

The results of the present study underline that in patients ≥80 years, overall frailty, which was already demonstrated to be strictly associated with in-hospital COVID-19 survival [13,15,16], is also associated with a poor long-term prognosis. As the long-term follow-up was considered, the analysis was adjusted for the expected survival for each five-year age group [28]. To exclude excess mortality due to COVID-19, we considered in the analysis the values reported in the year 2019. Interestingly, after adjusting for the potential confounders, the association between frailty and outcome was independent of most of the clinical factors considered, as well as the overall comorbidities, except for the non-autonomy in ADL. It could be speculated, however, that a comprehensive frailty evaluation includes, by definition, other known factors which have been associated with a poor COVID-19 prognosis in older adults, including dementia, relevant comorbidities, and delirium [3,12,13,14,15,16,17,18,19,35,36]. These results emphasize the usefulness of a comprehensive geriatric assessment in the risk stratification of patients ≥80 years affected by COVID-19. Moreover, given that most of the patients that survived the index hospitalization died within 1 year, special attention should be paid to the quality of life of the frailest adults in the post-discharge period. The impact of frailty on long-term prognosis and quality of life is not limited to hospitalization for COVID-19. In elderly patients, frailty is associated with a three-fold risk of recurrent hospitalization after admission for community-acquired pneumonia (CAP) [37]. At the same time, frailty is strongly associated with severe CAP and a higher 1-year mortality in elderly patients with CAP [38,39]. 

In the present study, the frailty was evaluated by the CSF which is widely used, efficient in emergency settings because of its simplicity [40], and already validated for the risk stratification of COVID-19 in older populations [14,16,17,19,33,41,42]. Due to the relevance of the frailty assessment by the CFS in the prognostic stratification of COVID-19, it is of the uttermost importance to determine the predictors of frailty worsening in older patients, particularly in adults ≥80 years, who can remain at high risk of COVID-19 even if vaccinated [43,44,45]. The present study revealed that no factor, among the clinical and demographic factors considered, was independently associated with a worsening in frailty in survivors >1 year since COVID-19 index hospitalization. A possible exception could be represented by the overall QOL estimated at the time of infection (Table 2). Although the analysis cannot provide conclusive clues on this point, it could be speculated that a reduced QOL could be associated with factors (i.e., reduced mobility, reduced self-care, and depression) that could be precipitated by the acute hospitalization for COVID-19, or by persistent post-COVID symptoms and induce an increase in the measured frailty. This is in line with previous reports [31], and indeed, the patients in our cohort experiencing an increase in the CFS had more persistent symptoms and a worse measured QOL compared to the controls (Table 2). However, the present analysis could not completely disclose the probable overlap and complex interactions among frailty, QOF, and post-COVID conditions.

A similar conclusion can be drawn for the worsening in overall QOL, as assessed by the 5-level EQ-5D. Most of the survivors in this group of older adults ≥80 years experienced a worsening in QOL after the index hospitalization (136/236, 57.6%). Interestingly, this effect was still evident 1 year after the event in survivors. These results are in line with previous reports [31,32,33,45]; however, the actual rate of patients experiencing a worsening in QOL in the present ≥80 years cohort was up to five-times higher compared to younger patients [45]. Interestingly, the reduction in the perceived QOL was not associated with the severity of the disease at the index hospitalization, nor with comorbidities or other relevant clinical conditions, but was about 30% less frequent in males (Table 3). At the same time, patients experiencing a worsening in perceived QOL had a higher number of persisting COVID symptoms, a higher number of hospital readmissions, and a higher estimated frailty at one year. Although our analysis could not reveal the underlying pathophysiological reasons for this, it could be speculated that, because the in-hospital mortality of the index hospitalization was much higher for males, this could have produced a selection of more fit male patients with an intrinsically better long-term prognosis. On the other hand, a gender-related difference in the odds for long COVID and reinfection from SARS-CoV2 was already found in previous research [46,47,48]. This is of great relevance in adults ≥80 years because, in the 8th and 9th decade, the female sex is more represented due to the higher life expectancy. However, because our analysis is limited to hospitalized patients, we cannot exclude a potential bias due to the disparities in sex distribution in both the elderly patients in nursing homes, which were hardly struck by the first COVID-19 waves, and the patients treated or deceased at home. For this reason, we cannot draw a definitive conclusion on the gender-related differences in the whole ≥80 years COVID-19 population.

### Study Limitations

Our study has some limitations that should be acknowledged. First, at the follow-up, we lost about 16% of the surviving patients. This number is not intrinsically high compared to similar cohorts; however, due to the reduced baseline expected survival of patients ≥80 years, we cannot omit that the lost patients could affect the overall survival analysis. Second, most of the post-COVID symptoms fluctuate in the affected population. For this reason, we cannot omit that a punctual evaluation of QOL and frailty could not be affected by these fluctuations. Finally, although all our patients had a hospitalization for COVID-19, we cannot exclude other concurrent causes of long-term worsening in QOL, frailty, and outcome.

## 6. Conclusions and Implications

In patients ≥80 years, the stratification of frailty by the CFS could predict the long-term survival of patients after hospitalization for COVID-19. This suggests particular attention to both the overall physical fitness and the quality of life of the frailest among the survivors. At the same time, hospitalization could produce a worsening in frailty itself by triggering a self-feeding mechanism between the increased risk of a new COVID-19 infection and a progressive increase in mortality risk.

Finally, our data show a clear effect on the long-term reduction in the perceived QOL among older COVID-19 survivors. This effect could be more pronounced in female patients with persisting symptoms, which could be the object of specific follow-up strategies and geriatric interventions.

## Figures and Tables

**Figure 1 jcm-11-05787-f001:**
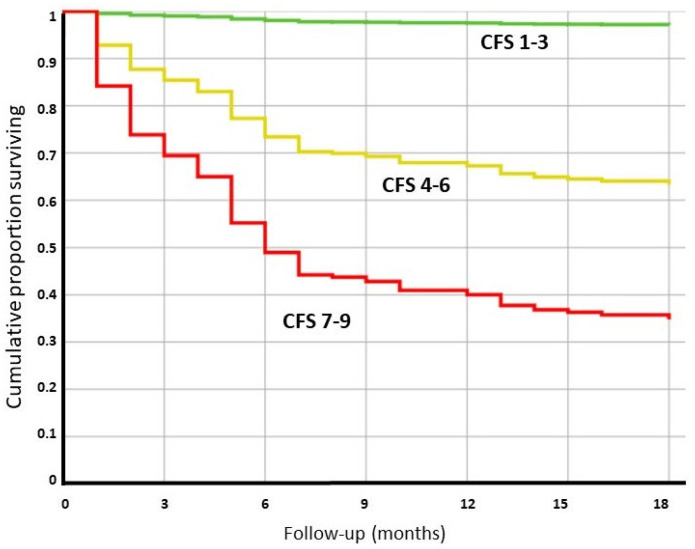
Cox regression analysis for patients with different frailty statuses (based on the clinical frailty scale) measured at the time of the index COVID-19 hospitalization. The analysis was adjusted for age, sex, life expectancy at 1 year, relevant clinical parameters, length of hospitalization, and comorbidities. The expected survival was based on the data obtained from the Italian Population Registry in 2019.

**Figure 2 jcm-11-05787-f002:**
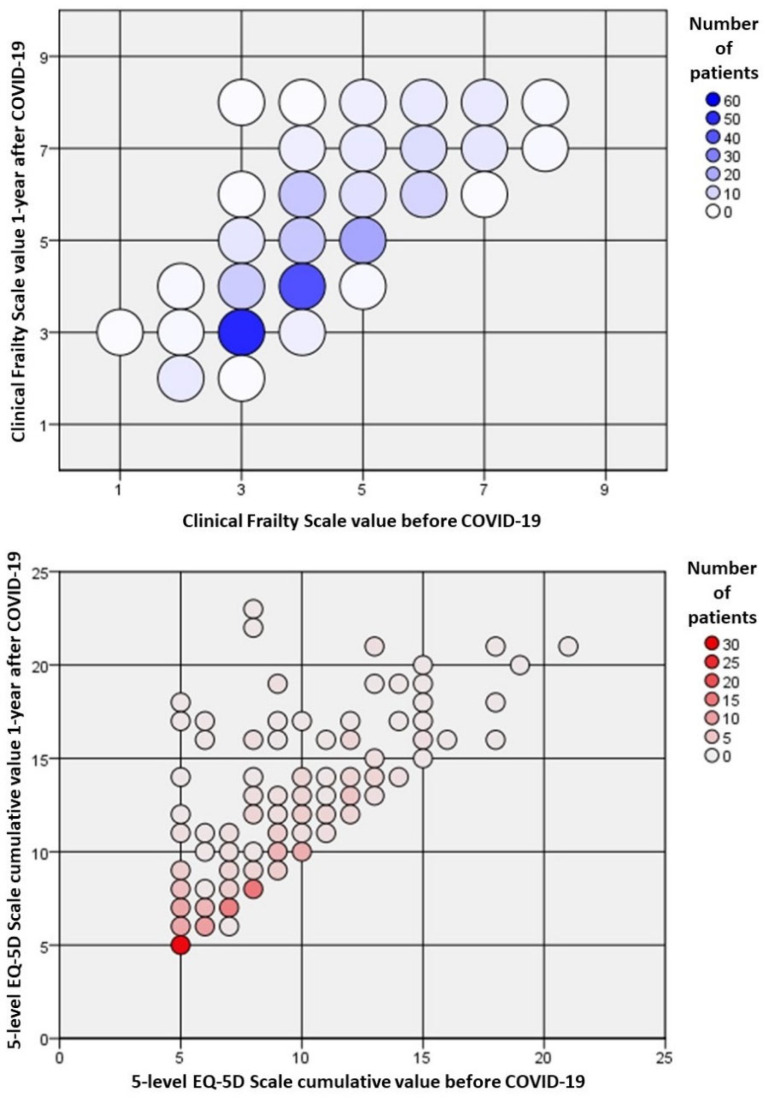
Relative changes in frailty and quality of life of survivors one year after the index hospitalization for COVID-19. Frailty was measured according to the clinical frailty scale (CFS), and the quality of life was assessed by the cumulative value of the five-level EQ-5D (EQ-5D-5L). Most of the patients had a worsening in frailty and the measured quality of life.

**Table 1 jcm-11-05787-t001:** Clinical characteristics of enrolled patients according to death at post-discharge follow-up. Adjusted hazard ratios were calculated by a multivariate Cox regression model (model chi-square = 119.586, *p* < 0.001; log-likelihood = 1394.384). Expected survival at 1 year was assessed based on data from the Italian Population registry in 2019. Proportions in the survived/deceased columns are reported as row percentages.

	All cases N 368	Survived N 236	Deceased N 132	*p* Value	Hazard Ratio [95% Confidence Interval]	
Age	85 [82, 89]	84 [81, 87]	87 [83, 91]	<0.01	1.08 [0.97, 1.21]	0.16
Age 80–85 years	177 (48.1%)	131 (74.0%)	46 (26.0%)			
Age 85–89 years	111 (30.2%)	73 (65.8%)	38 (34.2%)	<0.01		
Age 90–94 years	66 (17.9%)	29 (44.9%)	37 (56.1%)			
Age ≥ 95 years	14 (3.8%)	3 (21.4%)	11 (78.6%)			
Expected Survival/1 year	49.7% [49.7, 69.4]	69.4% [49.7, 69.4]	49.7% [29.6, 69.4]	<0.01	3.97 [0.25, 62.29]	0.33
Follow-up (months)	15 [6, 18]	17 [15, 18]	4 [2, 6]			
Sex (male)	163 (44.3%)	115 (70.6%)	48 (29.4%)	0.02	0.89 [0.62, 1.30]	0.58
*Frailty and self-reported quality of life before COVID*						
Clinical Frailty Scale (CFS)	5 [4, 6]	5 [4, 6]	6 [6, 7]	<0.01		
CFS 1–3	52 (14.4%)	50 (96.2%)	2 (3.8%)		*Reference category*	
CFS 4–6	237 (64.4%)	165 (69.6%)	72 (30.4%)	<0.01	4.91 [1.16, 20.70]	0.03
CFS 7–9	79 (21.5%)	21 (26.6%)	58 (73.4%)		6.61 [1.47, 29.80]	0.01
Resident in nursing home	92 (25.0%)	42 (45.7%)	50 (54.3%)	<0.01	1.06 [0.72, 1.55]	0.77
Autonomous in ADL (not)	213 (57.9%)	182 (85.4%)	31 (14.6%)	<0.01	3.55 [2.20, 5.78]	<0.01
EQ-5D-5L cumulative value	8 [5, 10]	9 [7, 13]	/	/		
*Clinical characteristics of the COVID-19 disease*						
PaO_2_/FiO_2_ at ED admission	295 [233, 357]	290 [233, 344]	304 [228, 376]	0.42		
NEWS at ED admission	5 [4, 7]	5 [4.75, 6.25]	5 [4, 8]	0.80		
NEWS > 5 at ED admission	16 (4.3%)	6 (37.5%)	10 (62.5%)	0.02	1.41 [0.73, 2.70]	0.29
Consolidation at chest X-ray	304 (82.6%)	201 (66.1%)	103 (33.9%)	0.08		
Delirium	34 (9.2%)	14 (41.2%)	20 (58.5%)	<0.01	1.41 [0.83, 2.72]	0.22
Mechanical ventilation	119 (32.3%)	81 (68.1%)	38 (31.9%)	0.28		
Length of hospital stay (days)	14.3 [8.5, 22.5]	13.1 [8.1, 22.2]	17.3 [10.0, 23.1]	0.02	1.00 [0.97, 1.01]	0.89
*Comorbidities*						
CCI	5 [4, 6]	5 [4, 6]	5 [4, 6]	<0.01		
Comorbidities ≥ 3	127 (34.5%)	69 (54.3%)	58 (45.7%)	<0.01	1.08 [0.75, 1.55]	0.68
Hypertension	162 (44.0%)	120 (74.1%)	42 (25.9%)	<0.01		
History of CAD	54 (14.7%)	37 (68.5%)	17 (31.5%)	0.47		
Congestive heart failure	56 (15.2%)	32 (57.1%)	24 (42.9%)	0.24		
Cerebrovascular disease	13 (3.5%)	6 (46.2%)	7 (53.8%)	0.17		
Dementia	73 (19.8%)	21 (53.8%)	52 (46.2%)	<0.01		
COPD	57 (15.5%)	36 (63.2%)	21 (36.8%)	0.87		
Diabetes	90 (24.5%)	53 (58.9%)	37 (41.1%)	0.23		
Chronic kidney disease	34 (9.2%)	19 (55.9%)	15 (44.1%)	0.29		
Malignancy	9 (2.4%)	3 (33.3%)	6 (66.7%)	0.07		

Abbreviations: EQ-5D-5L: 5 level EQ-5D; CCI: Charlson Comorbidity Index; ADL: activities of daily living; CAD: coronary artery disease; COPD: chronic obstructive pulmonary disease; NEWS: national early warning score.

**Table 2 jcm-11-05787-t002:** Clinical characteristics of enrolled patients according to increase in frailty as assessed by the Clinical Frailty Scale at 1-year follow-up. Logistic model chi-square = 16.167, *p* < 0.001; log-likelihood = 294.693. Expected survival at 1 year was calculated based on the data obtained from the Italian Registry of the population in 2019. Time was calculated from index ED admission for COVID. Proportions are reported as row percentages.

	Stable Frailty N 149	Increased Frailty N 87	*p* Value	Odds Ratio [95% Confidence Interval]	Multivariate *p*-Value
Age	84 [81, 86]	84 [81, 88]	0.06	1.04 [0.96, 1.12]	0.35
Age 80–85 years	87 (66.2%)	44 (6%)			
Age 85–89 years	44 (58.7%)	31 (41.3%)	0.37		
Age 90–94 years	13 (54.2%)	11 (45.8%)			
Age ≥ 95 years	5 (83.3%)	1 (16.7%)			
Expected Survival/1 year	69.4% [49.7, 69.4]	69.4% [49.7, 69.4]	0.30		
Sex (male)	81 (69.8%)	34 (30.2%)	0.02	0.61 [0.35, 1.07]	0.10
CFS pre-COVID	4 [3, 5]	4 [3, 5]	0.12	0.96 [0.75, 1.24]	0.77
Resident in nursing home	23 (54.8%)	19 (45.2%)	0.26		
Autonomous in ADL pre-COVID	121 (65.0%)	61 (35.0%)	0.05	0.51 [0.17, 1.50]	0.20
EQ-5D-5L before COVID	7 [5, 10]	9 [6.75, 12]	<0.01	1.12 [1.01, 1.24]	0.03
*Clinical characteristics of the COVID-19 disease*					
PaO_2_/FiO_2_ at ED admission	290 [259, 335]	290 [213, 359]	0.72		
NEWS at ED admission	5 [4.5, 6]	6 [4.5, 7]	0.36		
NEWS > 5 at ED admission	2 (33.3%)	4 (66.7%)	0.12		
Consolidation at chest X-ray	125 (61.5%)	76 (38.5%)	0.47		
Delirium	10 (71.4%)	4 (28.6%)	0.51		
Mechanical ventilation	55 (67.9%)	26 (32.1%)	0.27		
Length of hospital stay (days)	12.7 [7.4, 19.4]	14.0 [8.4, 27.3]	0.05		
*Comorbidities*					
CCI	5 [4, 6]	5 [4, 6]	0.73		
Comorbidities ≥ 3	47 (68.1%)	22 (31.9%)	0.31		
Hypertension	74 (61.7%)	46 (38.3%)	0.63		
History of CAD	27 (72.9%)	10 (27.1%)	0.18		
Congestive heart failure	20 (62.5%)	12 (37.5%)	0.94		
Cerebrovascular disease	3 (50.0%)	3 (50.0%)	0.67		
Dementia	15 (53.6%)	13 (46.4%)	0.26		
COPD	19 (52.8%)	17 (47.2%)	0.16		
Diabetes	34 (64.1%)	19 (35.8%)	0.86		
Chronic kidney disease	12 (63.1%)	7 (26.9%)	0.99		
Malignancy	1 (33.3%)	2 (66.7%)	0.28		
*Outcomes at Follow-up*					
Follow-up (months)	16 [15, 18]	17 [16, 18]	0.06		
CFS post-COVID	4 [3, 5]	6 [6, 7]	<0.01		
EQ-5D-5L 1 year after COVID	8 [6, 11]	12 [9, 16]	<0.01		
Persistent COVID symptoms	0 [0, 1]	0.5 [0, 1]	0.01		
Re-hospitalization <1 year	31 (63.2%)	18 (36.7%)	0.97		

Abbreviations: CFS: Clinical Frailty Scale; EQ-5D-5L: five-level EQ-5D; CCI: Charlson Comorbidity Index; ADL: activities of daily living; CAD: coronary artery disease; COPD: chronic obstructive pulmonary Disease; NEWS: national early warning score.

**Table 3 jcm-11-05787-t003:** Univariate and multivariate factors associated with the worsening of quality of life (QOL), as assessed by the EQ-5D-5L tool, in COVID-19 survivors at one-year follow-up. Pre-COVID clinical frailty scale (CFS) and 5-level EQ-5D cumulative values were forced into the multivariate Cox regression model. Expected survival at 1 year was calculated based on the data obtained from the Italian Registry of population in 2019. Time was calculated from index ED admission for COVID. Proportions are reported as row percentages.

	Stable QOL N 100	Worsened QOL N 136	*p* Value	Hazard Ratio [95% Confidence Interval]	Multivariate *p* Value
Age	83 [81, 86]	84 [81, 87]	0.11	1.01 [0.96, 1.06]	0.65
Age 80–85 years	62 (47.7%)	69 (52.3%)			
Age 85–89 years	24 (32.0%)	51 (68.05%)	0.03		
Age 90–94 years	9 (37.5%)	15 (62.5%)			
Age ≥ 95 years	5 (83.3%)	1 (16.7%)			
Expected Survival/1 year	69.4% [49.7, 69.4]	69.4% [49.7, 69.4]	0.22		
Sex (male)	60 (52.2%)	55 (47.8%)	<0.01	0.69 [0.48, 0.98]	0.04
CFS pre-COVID	4 [3, 5]	5 [4, 5]	0.03	1.01 [0.87, 1.17]	0.896
Resident in nursing home	16 (38.1%)	26 (71.9%)	0.59		
Autonomous in ADL pre-COVID	80 (43.9%)	102 (56.1%)	0.37		
EQ-5D-5L before COVID (cumulative)	7 [5, 9]	12 [8, 14]	0.03	1.00 [0.95, 1.05]	0.91
*Clinical characteristics of the COVID-19 disease*					
PaO_2_/FiO_2_ at ED admission	290 [233, 346]	288 [226, 346]	0.96		
NEWS at ED admission	5 [4, 6]	5.5 [5, 7]	0.39		
NEWS > 5 at ED admission	3 (50.0%)	3 (50.0%)	0.70		
Consolidation at chest X-ray	80 (39.8%)	121 (60.2%)	0.05		
Delirium	7 (50.0%)	7 (50.0%)	0.55		
Mechanical ventilation	33 (40.7%)	48 (59.3%)	0.71		
Length of hospital stay (days)	13.6 [9.0, 21.3]	12.4 [7.26, 22.4]	0.46		
*Comorbidities*					
CCI	5 [4, 6]	5 [4, 6]	0.78		
Comorbidities ≥ 3	32 (45.7%)	38 (54.3%)	0.43		
Hypertension	46 (58.3%)	74 (61.7%)	0.20		
History of CAD	20 (54.0%)	17 (66.0%)	0.12		
Congestive heart failure	13 (40.6%)	19 (59.4%)	0.83		
Cerebrovascular disease	3 (50.0%)	3 (50.0%)	0.70		
Dementia	11 (39.3%)	17 (60.7%)	0.72		
COPD	15 (41.7%)	21 (58.3%)	0.93		
Diabetes	21 (39.6%)	32 (60.4%)	0.64		
Chronic kidney disease	8 (42.1%)	11 (57.9%)	0.98		
Malignancy	0	3 (2.2%)	0.26		
*Outcomes at Follow-up*					
Total follow-up (months)	17 [15, 18.7]	17 [15, 18]	0.13		
CFS 1-year after COVID	4 [3, 5]	5 [4, 7]	<0.01		
EQ-5D-5L 1-year after COVID	8 [6, 11]	12 [9, 16]	<0.01		
Persistent COVID symptoms	0 [0, 1]	0 [0, 1]	<0.01		
Re-hospitalization <1 year	14 (40.0%)	35 (60.0%)	0.05		

Abbreviations: CCI: Charlson Comorbidity Index; ADL: activities of daily living; CAD: coronary artery disease; COPD: chronic obstructive pulmonary disease; NEWS: national early warning score.

## Data Availability

Data sharing is not applicable to this article.

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
