# Peer review of "Long-Term Effects of Hospitalization for COVID-19 on Frailty and Quality of Life in Older Adults ≥80 Years"

_jcm, 2022, doi:10.3390/jcm11195787_

Round 1

Reviewer 1 Report

Thank  your for offering me the opportunity to revise this interesting piece of work. In general speaking I consider this work appropiate according to the vulnerability of Covid-19 towards older population, the most affected by large by the pandemic. I specially like the focus on people older than 80 years old, instead of considering homogenously older people those older than 65 years old, as most of the articles did. The results are not entirely new but underscore once again the importance of frailty by itself in older population rather than particular comorbidities. The lack of association of most of comorbidities, including age, with the risk of death, decrease of life expectancy or quality of life in contrast with frailty scores is according to this observation.

I have one major consideration and some suggestions that may help to improve this article.

In my opinion the major weakness of the study is the lack of a control group of older population hospitalized by other non-Covid-19 related pathologies. The most appropiated would be Community Acquired Pneumonia. The study could not ascertain if the observed associations are specific of Covid-19 and at which level, or are a general trend of any hospitalization event among people older than 80 years. Authors could overcome this limitation by comparing their results with similar previous published studies during the pre-pandemic period.

Other minor comments:

Introduction: It is important to emphasize that during the first wave most of sever Covid-19 cases among older population occurred in long-term nursing homes. This may had introduced a certain selection biais since most of the residential population are females (around 70-75%). This may explain as well the striking observation that male gender was not correlated to bad outcome in contrast with previous reports among general population, including older people. Females were overrepresented and in general speaking older people hospitalized was biased since an important proportion died at home, nursing home premises and received paliative care. Surely, among this population males were overrepresented which may explain the ration 1:1 observed in the study cohort and may had introduced other biases. I suggest to the authors to discuss in deep these issues.

Material and methods: I strongly suggest to expand the description of the study center, the study region and study population covered by this facility. Howe many beds they have? It is a public or private facility? Which population covers? What is the proportion of people older than 80 years living in the study region?   It is a rural, urban area?

Results: If possible, mention how many enrolled patients came from the community and how many from long-care nursing homes. In this case, I would recommend an stratified analysis of both populations since the risk factors to acquire SARS-CoV-2 and the severity of the disease may be different between these two sub-groups of older people populations.

Table 1: I suggest to present the proportions of different variables within Survival and deceased columnes  as summatory rows, not in columns. This would make the differences between the two groups more clearer. It is not necessary to mention the Chi-square score or the log-likelihood value. Round the p values to one or two decimals.  For instance, instead of 0.282 use 0.3, for 0.067 use 0.07 and so one.

Figure 1: As far as I know Kaplan Meier projections could not be adjusted. They are just graphical or table projections from an observed cohorts.  I presume that authors refers to the Cox regression analysis to ascertain adjusted HR between different frailty groups.

Table 2: As mentioned before, I suggest to present the proportions of different variables among stable frailty and increased frailty   as summatory rows, not in columns. Round the p values to one or two decimals.

Discussion: I cannot completely agree with the statement that  according to the results“the frailest  adults should be probably included in strict post-discharge surveillance programs”. Of course, in general speaking frailest patients needs a more intensive care but considering the life expectancy that these patients have the focus should be put in the quality of care, and frequently, palliative and end-of-life care rather than intensive care. This includes social contact with relatives and friends and general well-being. The lack of social contact had a devastating effect among older population during the pandemic and the SARS-CoV-2 prevention among older population were misguided at this level.

Regarding the differences observed between males and females the authors correctly identified an in-hospital biais since males tend to died more frequently than females. Consider as well, as I mention above, the pre-hospitalization biais since males had higher chances to die before hospitalization. This may explain as well the gender balance 1:1 of the cohort under study. In my opinion, the conclusions made regarding the observed gender differences should be taken cautiously.

Author Response

Thank  your for offering me the opportunity to revise this interesting piece of work. In general speaking I consider this work appropiate according to the vulnerability of Covid-19 towards older population, the most affected by large by the pandemic. I specially like the focus on people older than 80 years old, instead of considering homogenously older people those older than 65 years old, as most of the articles did. The results are not entirely new but underscore once again the importance of frailty by itself in older population rather than particular comorbidities. The lack of association of most of comorbidities, including age, with the risk of death, decrease of life expectancy or quality of life in contrast with frailty scores is according to this observation.

  • We thank the reviewer for the time spent revising our paper, and for the thoughtful suggestions that indeed will consent us to ameliorate our work.

I have one major consideration and some suggestions that may help to improve this article.

 In my opinion the major weakness of the study is the lack of a control group of older population hospitalized by other non-Covid-19 related pathologies. The most appropiated would be Community Acquired Pneumonia. The study could not ascertain if the observed associations are specific of Covid-19 and at which level, or are a general trend of any hospitalization event among people older than 80 years. Authors could overcome this limitation by comparing their results with similar previous published studies during the pre-pandemic period.

  • We thank the reviewer for this precious suggestion. We agree with this observation, and we added to the manuscript a comparison to the results emerged from research in community acquired pneumonia elderly patients. We consequently added some references to the paper (37-39). See row 322-325 for the changes in the discussion.

Other minor comments:

Introduction: It is important to emphasize that during the first wave most of sever Covid-19 cases among older population occurred in long-term nursing homes. This may had introduced a certain selection biais since most of the residential population are females (around 70-75%). This may explain as well the striking observation that male gender was not correlated to bad outcome in contrast with previous reports among general population, including older people. Females were overrepresented and in general speaking older people hospitalized was biased since an important proportion died at home, nursing home premises and received paliative care. Surely, among this population males were overrepresented which may explain the ration 1:1 observed in the study cohort and may had introduced other biases. I suggest to the authors to discuss in deep these issues.

·         We thank the reviewer for this thoughtful observation. Undoubtedly a sex-related difference was observed both in short-term prognosis and in the likelihood of reinfection and long-covid. Furthermore, both the over-representation of women in nursing homes and factors related to home care capacity may have created a selective bias in hospitalized patients. However, since our analysis is limited to hospitalized patients and we have no data regarding the home-treated population, we cannot express a definitive opinion on sex-related differences in the overall over-80 population. This is now better clarified in the text (see row 359-363).

Material and methods: I strongly suggest to expand the description of the study center, the study region and study population covered by this facility. Howe many beds they have? It is a public or private facility? Which population covers? What is the proportion of people older than 80 years living in the study region?   It is a rural, urban area?

  • The suggested data are now better specified in the text (see row 107-108)

Results: If possible, mention how many enrolled patients came from the community and how many from long-care nursing homes. In this case, I would recommend an stratified analysis of both populations since the risk factors to acquire SARS-CoV-2 and the severity of the disease may be different between these two sub-groups of older people populations.

  • The residency in a nursing home was added to the analysis. Multivariate analysis was adjusted for this condition. However, since it showed a high co-linearity with “non-autonomous” it did not emerged as independent predictor for the overall survival.

Table 1: I suggest to present the proportions of different variables within Survival and deceased columnes  as summatory rows, not in columns. This would make the differences between the two groups more clearer. It is not necessary to mention the Chi-square score or the log-likelihood value. Round the p values to one or two decimals.  For instance, instead of 0.282 use 0.3, for 0.067 use 0.07 and so one.

  • To maintain the consistence among the data in the tables all the values were expressed as column percentages. However, we agree with the reviewer that row percentages will improve readability. The table 1 was changed according to this suggestion and the change was indicated in the table header.
  • The p values in all the tables were rounded to to 2 digits.

Figure 1: As far as I know Kaplan Meier projections could not be adjusted. They are just graphical or table projections from an observed cohorts.  I presume that authors refers to the Cox regression analysis to ascertain adjusted HR between different frailty groups.

  • We apologize for the typo. This has not been corrected in the figure legend.

Table 2: As mentioned before, I suggest to present the proportions of different variables among stable frailty and increased frailty   as summatory rows, not in columns. Round the p values to one or two decimals.

  • We agree with the reviewer that the table2 would improve in readability, and both table 2 and 3 were changed according to this suggestion.

Discussion: I cannot completely agree with the statement that  according to the results“the frailest  adults should be probably included in strict post-discharge surveillance programs”. Of course, in general speaking frailest patients needs a more intensive care but considering the life expectancy that these patients have the focus should be put in the quality of care, and frequently, palliative and end-of-life care rather than intensive care. This includes social contact with relatives and friends and general well-being. The lack of social contact had a devastating effect among older population during the pandemic and the SARS-CoV-2 prevention among older population were misguided at this level.

  • We strongly agree with the reviewer on this point. Actually the “post-discharge surveillance programs” were intended just in this way. This is now better specified in the text.

Regarding the differences observed between males and females the authors correctly identified an in-hospital biais since males tend to died more frequently than females. Consider as well, as I mention above, the pre-hospitalization biais since males had higher chances to die before hospitalization. This may explain as well the gender balance 1:1 of the cohort under study. In my opinion, the conclusions made regarding the observed gender differences should be taken cautiously.

·         As stated in the previous point a sex-related difference was observed both in short-term prognosis and in the likelihood of reinfection and long-covid in several studies. However, for the reasons already stated above, our data are insufficient to express a definitive opinion on sex-related differences in the overall over-80 population. This is now better clarified in the text (see row 359-363).

Reviewer 2 Report

This is an interesting study that analyzes the long-term effects of COVID-19 in a very vulnerable category - very old patients. Some findings are unusual, require confirmation in other studies, but are not without interest and have important practical meaning. While reading the manuscript, which is generally of good quality, there was one comment and several questions. Comment I don’t think it’s logical to group together those who required mechanical ventilation and patients with non-invasive respiratory support / high oxygen flow therapy Questions 1. How was dependency in ADL before SARS-CoV-2 infection assessed if the study was prospective and patients were enrolled in the ED of COVID-19 hospital? 2. And, similarly, when pre COVID-19 CFS and EQ-5D-5L scores were collected?  

Author Response

This is an interesting study that analyzes the long-term effects of COVID-19 in a very vulnerable category - very old patients. Some findings are unusual, require confirmation in other studies, but are not without interest and have important practical meaning. While reading the manuscript, which is generally of good quality, there was one comment and several questions. 

Comment I don’t think it’s logical to group together those who required mechanical ventilation and patients with non-invasive respiratory support / high oxygen flow therapy 

  • We thank the reviewer for this suggestion. Unfortunately, particularly in the first phase of the pandemic the patients were treated in sub-intensive and intensive units dedicate to COVID, and most patients shifted from one ventilation treatment to other during the course of the disease. Thus, we decided to group all these treatment in one single variable.

Questions 1. How was dependency in ADL before SARS-CoV-2 infection assessed if the study was prospective and patients were enrolled in the ED of COVID-19 hospital? 

  • We thank the reviewer for this question that consent us to clarify this point. As for any research on frailty and quality of life scale made in an acute setting the assessment is referred to the status of the patients before the acute event. This could be also evidenced by the official guidance and tips to administer the CFS scale (https://www.dal.ca/sites/gmr/our-tools/clinical-frailty-scale/cfs-guidance.html) that state in the first point: “If the person you are assessing is acutely unwell, score how they were 2 weeks ago, not how they are today.” Data were achieved by the patients, relative and health care providers. Since we have a dedicated geriatric unit in our emergency department the CFS and EQ-5D-5L scores were calculated for each elderly patients since 2019.
  1. And, similarly, when pre COVID-19 CFS and EQ-5D-5L scores were collected?  
  • Please see the point above.

Reviewer 3 Report

Due to lack of control group, the evidence was not so enough in the reduction of life quality reduction and mortality attributed to the infection with COVID-19.

From the table 1. It is really hard to understand the meaning of the figures in the table. Do the numbers in the table represent frequencies, means(average), or scores?

Why the authors selected the patient aged 80 years and over as the study objects rather than the patient aged 65 (or 60) years and older?

Author Response

Due to lack of control group, the evidence was not so enough in the reduction of life quality reduction and mortality attributed to the infection with COVID-19.

  • We thank the reviewer for this precious suggestion. We agree with this observation. Obviously, COVID-19 is not the only illness that could decrease the QOL and fitness in older patients. We included in the manuscript the results that emerged from similar research in community-acquired pneumonia in elderly patients. We consequently added some references to the paper (37-39). See rows 322-325 for the changes in the discussion.

From the table 1. It is really hard to understand the meaning of the figures in the table. Do the numbers in the table represent frequencies, means(average), or scores?

  • We thank the reviewer for this suggestion. The tables were changed to improve readability and the legend was improved to make it easier to understand the content.

Why the authors selected the patient aged 80 years and over as the study objects rather than the patient aged 65 (or 60) years and older?

Most of the research on elderly COVID patients focuses on patients ≥ 65 years including in this category the oldest in the 8th and 9th decade of life. Nevertheless, these latter patients are the most at risk and have different clinical and physical conditions, particularly regarding frailty. We believe that one of the strengths of our work is the focus on these very old patients that are the most rapidly increasing demographic group in western countries.

Round 2

Reviewer 1 Report

The questions raised during the first revision has been adequately addressed by the authors

Reviewer 3 Report

I am satisfied with the authors' effort in improving the quality of their manuscript despite they did not answer my previous questions completely.